# What Performative Contradiction Reveals: Plato's *Theaetetus* and *Gorgias* on Sophistry

**Robert Metcalf**

Department of Philosophy, University of Colorado, Denver, CO 80217, USA; robert.metcalf@ucdenver.edu

**Abstract:** Socrates' use of performative contradiction against sophistic theories is a recurrent motif in Plato's dialogues. In the case of Plato's *Theaetetus* and *Gorgias*, Socrates attempts to show that Protagoras' *homo mensura* doctrine and Gorgias' doctrine of the power of *logos* are each performatively contradicted by the underlying activity of philosophical dialogue. In the case of the *Theaetetus*, Socrates' strategy of performative contradiction hinges on Protagoras' failure to perform in the way that he theorized the sophist performing—namely, being able to change appearances through *logoi* (*Theaetetus* 166d–167d). In parallel fashion, Gorgias' account of the power of rhetoric is performatively contradicted by the orator's inability to prevail over Socrates, instead resorting to insincere responses to Socrates' questions in order to save face—a dialogical "performance" that ties directly to Socrates' portrait of Gorgianic rhetoric as a matter of pandering to the audience (*Gorgias* 460a–465a). Plato's aim in dramatizing these performative contradictions, I argue, is to illuminate both the proximity between Socrates and the great sophists, particularly with respect to Socrates' practice of *elenchos*, but also the distance between Socrates and the sophists in how they conceive of our situatedness within the world of human concerns.

**Keywords:** philosophy; sophistry; Plato; Socrates; Protagoras; Gorgias

## 1. Introduction

A recurrent motif in Plato's dialogues is the use of performative contradiction as a strategy in Socrates' refutation of sophistic theories. By "performative contradiction", I mean that form of refutation whereby a theory is shown to be false or is otherwise undermined by the underlying activity (*pragmateia*) in the context of which it is put forward for consideration. In this essay, I focus on the *Theaetetus*, in which Socrates lays out a performative contradiction of Protagoras' *homo mensura* doctrine, i.e., the teaching of Protagoras' most famous fragment, "Of all things the measure is man, of the things that are, that they are, and of things that are not, that they are not" (DK 80B1),[1] and the *Gorgias*, in which Socrates' refutation (*elenchos*) involves a performative contradiction of Gorgias' thesis that the orator "enslaves" the audience by means of persuasive *logos*. In what follows, I shall argue that what is revealed by performative contradiction in these dialogues is that Socrates' elenctic practice of philosophical dialogue is compatible neither with the *homo mensura* doctrine, nor with the Gorgianic doctrine of *logos*, according to which the power of *logos* is unidirectional, exercised by the orator upon the audience (see Wilburn in this Special Issue). Rather, *elenctic dialegesthai* presupposes that one may be wrong in one's beliefs, even wrong about what one believes, and presupposes further that one's power is measured by *logos*, not something that one exercises by "using" *logos* as an instrument or weapon. In this way, the recurring motif of performative contradiction carried out on sophistic thinking points us toward the philosophical proximity, but also the distance, between Plato's Socrates and these two great sophistic theorists.

However, to sustain our focus on performative contradiction vis à vis Protagoras and Gorgias, I will set aside two issues of scholarly debate that might otherwise overtake the discussion. The first issue is whether Protagoras of Abdera was a relativist in the way he is

presented in the *Theaetetus* (an issue that brings with it the question as to whether sophistic thought was generally "relativistic", in the way it is often characterized). The second issue is whether Gorgias of Leontini should be considered a sophist like Protagoras, Hippias, Prodicus and others. On the first issue, in what follows, I will not take up the question as to whether Protagoras' other extant fragments are consistent with the "relativism" expressed in DK 80B1.[2] On the second issue, I acknowledge up front that Gorgias is, in Plato's texts, sometimes called a sophist (e.g., in *Hippias Major* 282b and *Apology* 19d–20c) and sometimes portrayed as meaningfully distinct from the sophists and from sophistic thought (e.g., *Meno* 95b–c).[3] However, as I hope to show in what follows, the specific theory of sophistry articulated by the ventriloquized "Protagoras" in Plato's *Theaetetus*—namely, that the sophist is distinguished by his ability to make appearances change for people through his use of *logoi* (*Theaetetus* 166d–e)—applies as well to Gorgias as to any other candidate for "sophist", and this makes the parallel performative contradictions of Protagoras in the *Theaetetus* and Gorgias in the *Gorgias* worth examining together.[4]

An especially salient aspect of performative contradiction in Plato's texts is its robustly *performative*, "epideictic" character—something that Plato's texts are uniquely suited to showcase, since there is always, or nearly always, a rich dramatic and dialogical context within which theories like Protagoras' *homo mensura* and Gorgias' doctrine of *logos* are put forward for philosophical scrutiny.[5] At the same time, we see a significant difference between the dialogical and contextual situatedness of Socrates' *elenchos* of Protagoras in the *Theaetetus* and the very abstract logical analysis of Protagorean relativism in later thinkers, such as Aristotle, Sextus Empiricus, and modern commentators who have developed this analysis.[6] Consider Sextus' formulation of the "self-refutation" argument advanced against Protagoras: "One cannot say that every appearance (φαντασία) is true, because of its self-refutation (περιτροπή), as Democritus and Plato urged against Protagoras; for if every appearance is true, it will be true also, being in accordance with an appearance, that not every appearance is true, and thus it will become a falsehood that every appearance is true" (Sextus Empiricus, *Adversus mathematicos* 7: 389–90).[7] As Myles Burnyeat explains, the logic of the *peritrope* is something very specific. "Any refutation", he writes, "establishes the contradictory of what it refutes, but *peritrepein* tends particularly to be used of the special case where the thesis to be refuted itself serves as a premise for its own refutation . . . In such a case a thesis is turned around or reversed into its contradictory within the confines of a single inference."[8] We see this logic of the *peritrope* reflected also in Aristotle's *Metaphysics*, book *Gamma*, when he observes that such theories as Protagoras' *homo mensura* "destroy themselves [αὐτοὺς ἑαυτοὺς ἀναιρεῖν]; for he who says that everything is true makes the opposite theory true too, and therefore his own untrue [ὁ μὲν γὰρ πάντα ἀληθῆ λέγων καὶ τὸν ἐναντίον αὐτοῦ λόγον ἀληθῆ ποιεῖ, ὥστε τὸν ἑαυτοῦ οὐκ ἀληθῆ] (for the opposite theory says that his is not true [ὁ γὰρ ἐναντίος οὔ φησιν αὐτὸν ἀληθῆ]); and he who says that everything is false makes himself a liar [ὁ δὲ πάντα ψευδῆ καὶ αὐτὸς αὑτόν]" (1012b14–18).[9]

Yet, as nearly every commentator acknowledges, there are aspects of "self-refutation" involving Protagoras' *homo mensura* doctrine that reach beyond the narrow scope of the *peritrope*. Consider, for example, what is called the *pragmatic* self-refutation, whereby one puts forward a thesis while engaging in an activity that "contradicts" or performatively undermines the thesis.[10] One distinct kind of such pragmatic self-refutation is what Burnyeat has called *dialectical* self-refutation, wherein "it is the act of submitting a thesis for debate or maintaining it in the face of disagreement that causes its reversal and shows it up as false".[11] Additionally, as we shall see in what follows, there is, as further specifications of the performative contradiction dramatized in Plato, the genuinely performative or epideictic character of the *elenchos*. This is most prominent in Plato's *Gorgias*, since the dramatic setting involves Socrates' late arrival to Gorgias' "exhibition" (*epideixis*) before a crowd of admirers and would-be students, and yet Socrates' ensuing *elenchos* of Gorgias targets Gorgias' ability to 'perform' the thesis in question—namely, rhetoric's ability to

overpower and "enslave" the audience. Less prominently, perhaps, but nonetheless in parallel fashion, in the *Theaetetus*, Protagoras, as ventriloquized by Socrates, has to defend his own profession as a teacher while professing the *homo mensura* doctrine. Thus, the performative contradiction at stake in these texts is, properly understood, a pragmatic self-refutation, where the *pragmateia* at issue involves the epideictic performance of Protagoras and Gorgias in relation to the students or audience. Let us turn, then, to the details in Plato's portraits of these parallel instances of performative contradiction so as to clarify their philosophical significance.

## 2. Performative Contradiction in Plato's *Theaetetus*

To begin with the *Theaetetus*, consider the astonishing stretch of dialogue that occurs after Socrates portrays himself as a philosophical midwife who assists interlocutors in giving birth to something from out of their souls.[12] Insisting that the *logoi* never come from him, but rather always come from his interlocutor [οὐδεὶς τῶν λόγων ἐξέρχεται παρ᾽ ἐμοῦ ἀλλ᾽ ἀεὶ παρὰ τοῦ ἐμοὶ προσδιαλεγομένου], Socrates says that all he knows (ἐπίσταμαι), such as it is, is "how to take a *logos* from someone else, who is wise, and give it a measured reception [ὅσον λόγον παρ᾽ ἑτέρου σοφοῦ λαβεῖν καὶ ἀποδέξασθαι μετρίως]" (161b).[13] Socrates then states what he finds "astonishing" (ὃ θαυμάζω) about Protagoras:

> Well, I was delighted with his general statement of the theory that a thing is for any individual what it seems to him to be [ὡς τὸ δοκοῦν ἑκάστῳ τοῦτο καὶ ἔστιν]; but I was astonished [τεθαύμακα] at the way he began—namely, that he did not state at the beginning of the *Truth* that 'Pig is the measure of all things,' or 'Baboon' or some yet more outlandish creature with the power of perception [τι ἄλλο ἀτοπώτερον τῶν ἐχόντων αἴσθησιν] . . . It would have made it clear [ἐνδεικνύμενος] to us that, while we were standing astounded at his wisdom as though he were a god [ὥσπερ θεὸν ἐθαυμάζομεν ἐπὶ σοφίᾳ], he happened to be in reality no better positioned in his wisdom [ἐτύγχανεν ὢν εἰς φρόνησιν οὐδὲν βελτίων] than a tadpole—let alone any other human being [βατράχου γυρίνου, μὴ ὅτι ἄλλου του ἀνθρώπων]. Or what are we to say, Theodorus? (161c–d)[14]

Socrates is astonished that he and others are astounded at Protagoras' alleged wisdom, since the clear implication of his *homo mensura* doctrine is that Protagoras is no better positioned in wisdom—*sophia* and *phronêsis* being used interchangeably in this passage—than any other human being, and indeed no better positioned than any other creature having sense-perception, whether it be a baboon, a pig, or a tadpole.[15] Not only is Socrates piercingly funny in this passage critiquing Protagoras, but he shows what is involved in a performative contradiction: the thesis that a thing is for any individual what it seems to him to be is undermined by, or, at the very least, is shown to be out of harmony with, the activity in the context in which the thesis is put forward for consideration. This underlying activity (*pragmateia*), Socrates makes clear, is one that presupposes the differentiation between those who have wisdom and those without it, and presupposes, further, on the part of participants in *elenctic dialegesthai*, that things may not be as they seem to them to be.

This is further clarified by Socrates in the next part of the passage (the second of three parts of the passage that we will examine):

> If whatever the individual judges by means of perception is true for him [εἰ γὰρ δὴ ἑκάστῳ ἀληθὲς ἔσται ὃ ἂν δι᾽ αἰσθήσεως δοξάζῃ]; if no man can assess another's experience better than he [καὶ μήτε τὸ ἄλλου πάθος ἄλλος βέλτιον διακρινεῖ], or can claim authority to examine another man's judgment and see if it be right or wrong [μήτε τὴν δόξαν κυριώτερος ἔσται ἐπισκέψασθαι ἕτερος τὴν ἑτέρου ὀρθὴ ἢ ψευδής]; if, as we have repeatedly said, only the individual himself can judge of his own world [αὐτὸς τὰ αὑτοῦ ἕκαστος μόνος δοξάσει], and what he judges is always true and correct [ταῦτα δὲ πάντα ὀρθὰ καὶ ἀληθῆ]: how could it ever be, my friend, that Protagoras was a wise man [σοφός], so wise as to think himself fit to be the teacher of other men [ὥστε καὶ ἄλλων διδάσκαλος ἀξιοῦσθαι

δικαίως] and worth large fees; while we, in comparison with him the ignorant ones [ἀμαθέστεροί], needed to go and sit at his feet—we who are ourselves each the measure of his own wisdom [ἡμῖν . . . μέτρῳ ὄντι αὐτῷ ἑκάστῳ τῆς αὐτοῦ σοφίας]? (161d–e)

Here, Socrates details further the incompatibility between Protagoras' *homo mensura* doctrine and the underlying activity in the context in which it is put forward. On the one side, the *homo mensura* doctrine has it that whatever each individual judges by means of perception (ὃ ἂν δι᾽ αἰσθήσεως δοξάζῃ) is true for him, and each individual is the only judge of his own world (αὐτὸς τὰ αὑτοῦ ἕκαστος μόνος δοξάσει). Yet, on the other side, presupposed by the underlying activity of philosophical dialogue, is the effort to examine another man's judgment and see if it is right or wrong (τὴν δόξαν . . . ἐπισκέψασθαι ἕτερος τὴν ἑτέρου ὀρθὴ ἢ ψευδής) so as to distinguish the truly wise person (σοφός), the one worthy of being a teacher of others (ἄλλων διδάσκαλος ἀξιοῦσθαι). If, as Socrates says at the end, each of us is the measure of our own wisdom, the very activity of examining and critiquing the opinion (*doxas*) of others is rendered meaningless and indeed comical on account of its forgetfulness of itself while praising Protagoras' *Truth*.[16]

In the final stretch of the passage in question, Socrates concludes by underscoring the character of the performative contradiction, but also by raising the question of bad faith on the part of Protagoras:

Can we avoid the conclusion that Protagoras was just playing to the crowd [δημούμενον] when he said this? I say nothing about my own case and my art of midwifery and how ridiculous we look [ὅσον γέλωτα ὀφλισκάνομεν]. So too, the whole business of philosophical discussion [καὶ σύμπασα ἡ τοῦ διαλέγεσθαι πραγματεία], examining and trying to refute each other's appearances and judgments [τὸ γὰρ ἐπισκοπεῖν καὶ ἐπιχειρεῖν ἐλέγχειν τὰς ἀλλήλων φαντασίας τε καὶ δόξας], when each person's are correct [ὀρθὰς ἑκάστου οὔσας]—this is surely an extremely tiresome piece of nonsense [φλυαρία], if the *Truth* of Protagoras is true, and not merely an oracle speaking in jest from the impenetrable sanctuary of the book [ἀλλὰ μὴ παίζουσα ἐκ τοῦ ἀδύτου τῆς βίβλου ἐφθέγξατο]. (161e–162a)

The *pragmateia* to which Socrates directs our attention when he speaks of the "whole business of philosophical discussion [σύμπασα ἡ τοῦ διαλέγεσθαι πραγματεία]" is the activity in the context of which Protagoras' *homo mensura* doctrine suffers "pragmatic self-refutation", and the *pragmateia* operative here is quite complicated: examining each other's appearances and opinions and trying to refute them. Later in the dialogue, Socrates details this activity somewhat further when he characterizes the philosopher devoting his life to philosophy (ἐν φιλοσοφίᾳ διάγουσι) as a matter of engaging in inquiry and involving himself in the effort to investigate (ζητεῖ τε καὶ πράγματ᾽ ἔχει διερευνώμενος) (174b).[17] Yet, as Socrates makes the point, the *homo mensura* doctrine precludes the very possibility of being able to critique each other's appearances and opinions, indeed Socratic midwifery would become ridiculous nonsense, for *elenctic dialegesthai* and Socrates' maieutic art require a shared venue for the assessment of one another's opinions (implicit in Socrates' conception of "seeking in common" [*zêtein koinê*])—and precisely this is ruled out by Protagoras' idea that each human being is self-sufficient in wisdom. When Socrates glosses his maieutic art as receiving an account from another, someone who has wisdom *in a measured* way (μετρίως) (161b), he is, in effect, contrasting a practice that differentiates between those who have wisdom and those who do not from the Protagorean doctrine according to which each human being is the measure.

But the second point made by Socrates in this passage is the possibility that Protagoras is playing to the crowd, engaging in *dêmêgoria*.[18] In other words, Socrates suggests that the *homo mensura* doctrine is what the crowd wants to hear—it gratifies their appetites, which are, after all, what an orator must satisfy in his effort to persuade them. Is Protagorean relativism, we wonder, the required ideology of democratic culture? Socrates reiterates this suggested point when he goes on to ask Theaetetus if he is not astonished at suddenly

finding himself, according to the Protagorean measure (τὸ Πρωταγόρειον μέτρον), "the equal of any man or even a god in wisdom [μηδὲν χείρων εἰς σοφίαν ὁτουοῦν ἀνθρώπων ἢ καὶ θεῶν]" (162c). When Theaetetus replies that he was convinced that the *homo mensura* doctrine was a sound one upon first hearing it and is indeed astonished (πάνυ θαυμάζω) by it, Socrates says, "Yes, because you are young, dear lad; and so you lend a ready ear to mob-oratory [νέος γὰρ εἶ, ὦ φίλε παῖ: τῆς οὖν δημηγορίας ὀξέως ὑπακούεις] . . . and say whatever is likely to be acceptable to the many [καὶ ἃ οἱ πολλοὶ ἂν ἀποδέχοιντο ἀκούοντες, λέγετε ταῦτα]" (162d–e). A measure of equality is put forward in the *homo mensura* doctrine, since each human being is made equal in the eyes of Protagoras, but with this equality and its ruling out epistemic differences between human beings there is also what we might call *undifferentiation*, where no one is singled out for distinction, and where, therefore, human beings in the plural represent an undifferentiated mass.[19] In this connection, it is impossible not to think immediately of the *Republic*'s portrait of the audience targeted by those who are called "sophists"—namely, the multitude, whose desires are communicated to the sophistic orator not through distinct voices but, as it were, through the murmurings of a great beast:

> Each of these private teachers who work for pay, whom the politicians call sophists and regard as their rivals, inculcates nothing else than these opinions of the multitude which they opine when they are assembled [μὴ ἄλλα παιδεύειν ἢ ταῦτα τὰ τῶν πολλῶν δόγματα, ἃ δοξάζουσιν ὅταν ἀθροισθῶσιν] and calls this knowledge wisdom [καὶ σοφίαν ταύτην καλεῖν]. It is as if a man were acquiring the knowledge of the humors and desires of a great strong beast which he had in his keeping, how it is to be approached and touched, and when and by what things it is made most savage or gentle, yes, and the several sounds [φωνὰς] it is wont to utter [φθέγγεσθαι] on the occasion of each . . . knowing nothing in reality about which of these opinions and desires is honorable or base, good or evil, just or unjust, but should apply all these terms to the judgements of the great beast, calling the things that pleased it good, and the things that vexed it bad, having no other account to render of them . . . Does it not seem to you that such a one would be a strange educator [ἄτοπος ἂν σοι δοκεῖ εἶναι παιδευτής]? (493a–c).[20]

When Socrates subsequently presents Protagoras defending himself against the criticism in the *Theaetetus*, it is clear that Protagoras is taking pains to finesse the *homo mensura* doctrine in such a way that it will not performatively run afoul of the underlying *pragmateia* of *dialegesthai*:[21]

> I take my stand on the truth being as I have written it. Each one of us is the measure both of what is and of what is not [μέτρον γὰρ ἕκαστον ἡμῶν εἶναι τῶν τε ὄντων καὶ μή]; but there are countless differences between men just for this very reason, that different things both are and appear [ἔστι τε καὶ φαίνεται] to be to different subjects. I certainly do not deny the existence of both wisdom and wise men—far from it. But the man that I call wise is the man who in any case where bad things both appear and are for one of us, works a change and makes good things appear and be for him [μεταβάλλων ποιήσῃ ἀγαθὰ φαίνεσθαί τε καὶ εἶναι] . . . What we have to do is to make a change from the one to the other, because the other state is *better*. In education too what we have to do is to change a worse state into a better state; only whereas the doctor brings about the change by the use of drugs, the sophist does it by the use of words [ἀλλ᾽ ὁ μὲν ἰατρὸς φαρμάκοις μεταβάλλει, ὁ δὲ σοφιστὴς λόγοις]. What never happens is that a man who judges what is false is made to judge what is true. For it is impossible to judge what is not, or to judge anything other than what one is immediately experiencing [παρ᾽ ἃ ἂν πάσχῃ]; and what one is immediately experiencing is always true [ταῦτα δὲ ἀεὶ ἀληθῆ] . . . Similarly, the wise and good orator is the man who makes wholesome things seem just [δίκαια δοκεῖν] to a city instead of

pernicious ones. Whatever in any city is regarded as just and admirable *is* just and admirable; but the wise man replaces each pernicious convention by a wholesome one [χρηστὰ], making this both be and seem so [ἐποίησεν εἶναι καὶ δοκεῖν]. By the same token the sophist who is able to educate his pupils along these lines is a wise man, and is worth his large fees to them [κατὰ δὲ τὸν αὐτὸν λόγον καὶ ὁ σοφιστὴς τοὺς παιδευομένους οὕτω δυνάμενος παιδαγωγεῖν σοφός τε καὶ ἄξιος πολλῶν χρημάτων τοῖς παιδευθεῖσιν]. In this way we are enabled to hold both that some men are wiser than others, and also that no man judges what is false [καὶ οὕτω σοφώτεροί τέ εἰσιν ἕτεροι ἑτέρων καὶ οὐδεὶς ψευδῆ δοξάζει]. And you, too, whether you like it or not, must put up with being a 'measure.' (166d–67d)

One thing particularly fascinating about the defense that Socrates' Protagoras mounts here is its theory of sophistry. The sophist wields a kind of power and wisdom that differentiates him from others in the *polis* in his capacity to make things appear through the use of words, *logoi*. The power of sophistry is made evident within a theater of seeming and appearing, *dokein* and *phainesthai*. In this way, the Protagoras fashioned by Socrates in the *Theaetetus* seems to bring together the figures of Protagoras and Gorgias.[22] For Protagoras, the theorist of relativism and the professional teacher of excellence through the use of words, is here presented in closest proximity to Gorgias the theorist of the daimonic power of *logos*, carrying out on the souls of the audience something analogous to what the physician carries out on bodies through the use of *pharmaka*—an analogy explicitly drawn by both Protagoras in the *Theaetetus* and Gorgias in the *Gorgias*.[23]

Now, does the defense proffered by Protagoras here save sophistry from the performative contradiction? The general contours of Socrates' gambit in response to Protagoras is evident in Plato's *Cratylus*, in which the performative contradiction is not dramatized as it is in the interactions between the characters in *Theaetetus* and *Gorgias* but instead is sketched out theoretically in succinct fashion. Notice that Socrates frames his question about the *homo mensura* doctrine to Hermogenes in *Cratylus* 385e–386a using the verbs *phainesthai* and *dokein*, as he does again at beginning of 386b. Socrates asks Hermogenes whether, if Protagoras is right, and "all things are to each person as they seem to him [τὸ οἷα ἂν δοκῇ ἑκάστῳ τοιαῦτα καὶ εἶναι]", it is possible for some of us to be wise (τοὺς μὲν ἡμῶν φρονίμους εἶναι) and some foolish (τοὺς δὲ ἄφρονας)—a possibility that Hermogenes understands to be precluded by the *homo mensura* doctrine (386c–d). Yet, as Socrates shows, if it seemed very strongly to Hermogenes or another interlocutor (ταῦτά γε . . . σοὶ πάνυ δοκεῖ) that Protagoras is not right in holding that whatever seems to each person is really true to him (δοκῇ ἑκάστῳ ἀληθῆ ἔσται), or if it seemed to such an interlocutor that, in truth, one individual is wiser than another (ὁ ἕτερος τοῦ ἑτέρου φρονιμώτερος), what could Protagoras possibly say in rebuttal? The argumentative gambit here, in other words, is to turn the *homo mensura* doctrine against itself, to use the context of *dokein*/*phainesthai* against the thinker who appeals to it as the arbiter or measure of truth.[24] The only possible response on the part of the relativist, as the Protagoras of the *Theaetetus* rightly apprehends, is to change the appearances or how things seem on the part of the interlocutor—his success at which would be a testament to his wielding sophistic power.

Thus, if performative contradiction is going to work against the *homo mensura* doctrine, it will hinge on the failure of Protagoras to perform in the way that he has theorized the sophist performing. Can Protagoras succeed in making it appear to Socrates and Theodorus that each human being is the measure, each human being self-sufficient with respect to wisdom? Dramatically, the dialogue between Socrates and Protagoras has become an *agôn* in which either the one claiming to have sophistic wisdom performatively contradicts his own theory, thereby making himself ridiculous, or Socrates himself turns into a laughing-stock in his contestation of Protagoras. At this point Socrates asks Theodorus whether he noticed how Protagoras, in responding to their contestation of his doctrine (ἀγωνιζοίμεθα εἰς τὰ ἑαυτοῦ) was disparaging Socrates' method of argument as an "amusing game [χαριεντισμόν τινα]" while treating his own doctrine solemnly as a matter of

seriousness (σπουδάσαι) (168d), and Socrates subsequently warns Theodorus that they must not unconsciously slip into some childish form of argument (μή που παιδικόν τι λάθωμεν εἶδος τῶν λόγων) (169c–d). Socrates retraces with Theodorus the question at issue in assessing his reformulated doctrine: namely, whether they were "correct or not in being displeased [ὀρθῶς ἢ οὐκ ὀρθῶς ἐδυσχεραίνομεν] and holding it against the *logos* which makes each individual self-sufficient with respect to wisdom [ἐπιτιμῶντες τῷ λόγῳ ὅτι αὐτάρκη ἕκαστον εἰς φρόνησιν ἐποίει]" (169d). Socrates formulates his reply to Protagoras as follows:

> Well, then, Protagoras, we too are expressing how it seems to a man—I might say, to all men [καὶ ἡμεῖς ἀνθρώπου, μᾶλλον δὲ πάντων ἀνθρώπων δόξας λέγομεν]—when we say that there is no one in the world who doesn't believe that in some matters he is wiser than other men, while in other matters they are wiser than he [καὶ φαμὲν οὐδένα ὄντινα οὐ τὰ μὲν αὐτὸν ἡγεῖσθαι τῶν ἄλλων σοφώτερον, τὰ δὲ ἄλλους ἑαυτοῦ]. In emergencies—if at no other time—you see this belief. When they are in distress, on the battlefield, or in sickness or in a storm at sea, all men turn to their leaders in each sphere as to God, and look to them for salvation because they are superior in precisely this one thing—knowledge [οὐκ ἄλλῳ τῳ διαφέροντας ἢ τῷ εἰδέναι]. And all the world of human beings is full of people seeking teachers and masters for themselves and for other living creatures and for works/deeds [καὶ πάντα που μεστὰ τἀνθρώπινα ζητούντων διδασκάλους τε καὶ ἄρχοντας ἑαυτῶν τε καὶ τῶν ἄλλων ζῴων τῶν τε ἐργασιῶν]; and on the other side, there are those who believe that they are competent to teach and competent to lead [οἰομένων τε αὖ ἱκανῶν μὲν διδάσκειν, ἱκανῶν δὲ ἄρχειν εἶναι]. In all these cases, what else can we say but that men do believe in the existence of both wisdom and ignorance among themselves [αὐτοὺς τοὺς ἀνθρώπους ἡγεῖσθαι σοφίαν καὶ ἀμαθίαν εἶναι παρὰ σφίσιν]? . . . And they believe that wisdom is true thinking, while ignorance is a matter of false judgment [οὐκοῦν τὴν μὲν σοφίαν ἀληθῆ διάνοιαν ἡγοῦνται, τὴν δὲ ἀμαθίαν ψευδῆ δόξαν]? [Theodorus answers affirmatively.] What then, Protagoras, are we to make of your argument [τί οὖν, ὦ Πρωταγόρα, χρησόμεθα τῷ λόγῳ;]? Are we to say that all men, on every occasion, judge what is true? Or that they judge sometimes truly and sometimes falsely [ἢ τοτὲ μὲν ἀληθῆ, τοτὲ δὲ ψευδῆ;]? Whichever we say, it comes to the same thing, namely, that men do not always judge what is true; that human judgments are both true and false [μὴ ἀεὶ ἀληθῆ ἀλλ᾽ ἀμφότερα αὐτοὺς δοξάζειν]. (170a–c)

This passage is remarkable for so many reasons. One of them is the fact that Socrates has, in effect, fleshed out the context of human life in which knowing (εἰδέναι) and wisdom (σοφία) are urgent matters: namely, when human beings are in distress, when they are in times of war, in times of sickness or when they are threatened by storms at sea. We should keep in mind that in the outer dramatic frame of the dialogue, Theaetetus—now a grown man of some accomplishment—has been wounded in battle and is dying of dysentery (142a–b).[25] The remembered conversation between him and Socrates on knowledge is brought to mind by his comrades in the face of the very exigencies in human life that make knowledge a matter of urgency (142c–d). But there is more to the context than that. Not only are knowledge and wisdom a matter of urgency because of distress, disease, war and natural disaster, it is also the case that human beings have radically different epistemic 'stations' in relation to one another. Human beings do not always judge truly; false judgments are rampant, and it is precisely because of this fact that there is constantly underway in human life a seeking after those who have the knowledge competent to address the issues that plague human beings. There are those who suppose themselves competent to lead and put themselves forward as teachers, and those who recognize themselves as needing a teacher or leader to guide them. Thus, the underlying context and activity within which Protagoras puts himself forward as someone with wisdom sufficient to teach others—the *pragmateia* within which sophistry emerges as something sought after

as a remedy—is such that human beings are recognized as not being self-sufficient with respect to wisdom. It is this *pragmateia* that the *homo mensura* is radically out of step with, and the performative contradiction at issue here in the *Theaetetus* is the revealing of this comical inconsistency.

To amplify the point here, consider that Socrates, in the stretch of dialogue following the passage just quoted, brings to their attention the displeasure, non-agreement, and contestation involved in the *pragmateia* in question. Socrates begins by asking Theodorus, "Would you, would anyone of the school of Protagoras, be prepared to contend [εἰ ἐθέλοι ἄν τις τῶν ἀμφὶ Πρωταγόραν ἢ σὺ αὐτὸς διαμάχεσθαι] that no one ever thinks his neighbor is ignorant or judging falsely [ὡς οὐδεὶς ἡγεῖται ἕτερος ἕτερον ἀμαθῆ τε εἶναι καὶ ψευδῆ δοξάζειν]?", to which Theodorus replies, "No that's not a thing one could believe, Socrates [ἀλλ᾽ ἄπιστον, ὦ Σώκρατες]" (170c). Immediately, Socrates reiterates the point that this unbelievable (*apiston*) implication follows necessarily from Protagoras' account that man is the measure of all things (καὶ μὴν εἰς τοῦτό γε ἀνάγκης ὁ λόγος ἥκει ὁ πάντων χρημάτων μέτρον ἄνθρωπον λέγων) (170d). Socrates asks if Theodorus or any other ally of Protagoras would be willing to contest (*diamachesthai*) the necessary implication of the *homo mensura* doctrine when it would make a mockery of all contesting-with-one-another-in-dialogue. Socrates then asks whether, quite the opposite of granting that the opinions and judgments of others are always true (ἀεὶ σὲ κρίνομεν ἀληθῆ δοξάζειν), "Isn't it the case that myriads of people on each occasion oppose their opinions to yours, believing that your judgment and belief are false [ἢ μυρίοι ἑκάστοτέ σοι μάχονται ἀντιδοξάζοντες, ἡγούμενοι ψευδῆ κρίνειν τε καὶ οἴεσθαι]?" (170d). Theodorus concedes this, adding that these individuals give him all the trouble (*pragmata*) that is humanly possible (170d). Socrates then concludes his *elenchos* of Protagoras and the *homo mensura* doctrine as follows:

> It will be disputed, then, by everyone, beginning with Protagoras [ἐξ ἁπάντων ἄρα ἀπὸ Πρωταγόρου ἀρξαμένων ἀμφισβητήσεται]—or rather, it will be admitted by him [ὑπό γε ἐκείνου ὁμολογήσεται], when he grants to the person who contradicts him that he judges truly [ὅταν τῷ τἀναντία λέγοντι συγχωρῇ ἀληθῆ αὐτὸν δοξάζειν]—when he does that, even Protagoras himself will be granting that neither dog nor the 'man in the street' is the measure of anything at all which he has not learned [τότε καὶ ὁ Πρωταγόρας αὐτὸς συγχωρήσεται μήτε κύνα μήτε τὸν ἐπιτυχόντα ἄνθρωπον μέτρον εἶναι μηδὲ περὶ ἑνὸς οὗ ἂν μὴ μάθῃ] . . . Then since it is disputed by everyone [ἀμφισβητεῖται ὑπὸ πάντων], the Truth of Protagoras is not true for anyone at all, not even for himself [οὐδενὶ ἂν εἴη ἡ Πρωταγόρου Ἀλήθεια ἀληθής, οὔτε τινὶ ἄλλῳ οὔτ᾽ αὐτῷ ἐκείνῳ]? (171b–c)

Theodorus protests that they are running down his friend too hard, at which point Socrates imagines Protagoras reemerging to carry out his own *elenchos* on them (171c–d).

The point of retracing these argumentative steps in as detailed a way as we have here is to keep track of the phenomena that Socrates has given attention to as the dialogical context within which Protagoras' *homo mensura* doctrine is put forward: namely, the life-and-death exigencies of human life that make the difference between wisdom and ignorance a matter of urgency, and more generally, the disagreements with one another, the attempted refutations of one another and challenges to one another's judgments, all of which frame the "situatedness" of sophistic theories like Protagoras' *homo mensura* doctrine. The performative contradiction Socrates reveals in what Protagoras has put forward in the *Theaetetus* is thus a matter of the *homo mensura* doctrine being performatively out of step with the very conditions for the possibility of that theory—unless, as Protagoras contends, he can wield the sophistic power he has theorized in *making it appear* to Socrates and Theodorus and others that the *homo mensura* doctrine withstands the refutation advanced by Socrates. Here, we see the fully epideictic character of the performative contradiction, for the 'debate' between Socrates and Protagoras as ventriloquilized by Socrates is itself an exhibition either of Protagoras' sophistic power in making-appear-through-*logoi* or his failure to wield such power. This epideictic character of the performative contradiction dramatized in *Theaetetus* invites comparison with the *epideixis* dramatized in Plato's *Gorgias*, which involves not a

ventriloquized sophistic theorist but Gorgias himself, whose exhibition before admiring prospective students is the event that Socrates has arrived at too late to witness.

### 3. Performative Contradiction in Plato's *Gorgias*

If "Platonic irony" is, as Drew Hyland has argued, the writing of the dialogues in such a way that "things are not always what they seem and what is said is not necessarily what is to be believed", (Hyland 2008, p. 92) then no dialogue displays such irony more clearly than Plato's *Gorgias*. As indicated previously, the dramatic setting of the dialogue has it that Socrates has come to witness the Sicilian orator's *epideixis*, but on account of his late arrival he must ask Gorgias to engage with him in *dialegesthai* about what his power and *technê* amount to (447a–c).[26] Here, at the beginning of the dialogue, Socrates presents himself as though he is unaware that rhetoric is the power and alleged *technê* taught by Gorgias, and as though his questioning of Gorgias were a good faith effort to learn something for the first time. Only later, after Polus takes over as the principal interlocutor, do readers realize that in fact Socrates has come to Gorgias' event with an entire speech about rhetoric at the ready (a speech whose taxonomy of rhetoric will be referred to throughout the rest of the dialogue [e.g., 500a–501c; 513d–e; 517a–522c; 527b–c]) and that its purpose is to overturn Gorgias' account of rhetoric.[27] In other words, Socrates' questioning of Gorgias is aimed from the outset at contesting and overturning the orator's claims to power and *technê*, and indeed, his ensuing account of rhetoric as pandering (κολακεία) makes it clear that Socrates' aim is to use Gorgias' own concepts of power, *logos* and *psychē* to discredit Gorgianic rhetoric.[28] Here, we find a revealing parallel with the *Theaetetus* in how Socrates intends to carry out an *elenchos* of Gorgias through performative contradiction: namely, by showing how Gorgias' account of *logos* as a "powerful lord" (δυνάστης μέγας) is out of harmony with his own performance or exhibition of rhetoric and with the underlying activity in the context of which it is put forward as an account.

Early in Socrates' questioning, Gorgias concedes that, unlike other *technai*, where the knowledge (ἐπιστήμη) in question is about something tangible, as in "handiwork" (χειρουργία), Gorgianic rhetoric is exclusively about *logoi* (449d).[29] More specifically, Gorgias says, rhetoric makes men able to speak (λέγειν γε ποιεῖ δυνατούς) as well as understand (φρονεῖν) the things about which they speak (449e), and its "entire activity and efficacy is by way of speeches [πᾶσα ἡ πρᾶξις καὶ ἡ κύρωσις διὰ λόγων ἐστίν]" (450b–c). When asked by Socrates what rhetorical *logoi* are concerned with (451d), Gorgias responds, "The greatest of human affairs, Socrates, and the best [τὰ μέγιστα τῶν ἀνθρωπείων πραγμάτων . . . καὶ ἄριστα]" (451d). He then goes on to identify this "greatest good" as a cause not merely of freedom for human beings but also of rule (ἄρχειν) in each city:[30]

> I call it the ability to persuade with speeches [τὸ πείθειν ἔγωγ᾽ οἷόν τ᾽ εἶναι τοῖς λόγοις] either judges in the lawcourts or statesmen in the council-chamber or the commons in the Assembly or an audience at any other meeting that may be held on public affairs. And I tell you that by virtue of this power you will have the doctor as your slave [ἐν ταύτῃ τῇ δυνάμει δοῦλον μὲν ἕξεις τὸν ἰατρόν], and the trainer as your slave; your money-getter will turn out to be making money not for himself, but for another—in fact, for you, who are able to speak and persuade the multitude [ἀλλὰ σοὶ τῷ δυναμένῳ λέγειν καὶ πείθειν τὰ πλήθη]. (452e)

Socrates replies, "You seem now to me, Gorgias, to have come very near to making clear what you think the art of rhetoric is, and if I am following you at all, you say that rhetoric is an artificer of persuasion [πειθοῦς δημιουργός ἐστιν ἡ ῥητορική], and therein has its whole business and consummation" (452e–453a). Socrates then asks Gorgias whether rhetoric has any other "power" (δύνασθαι) beyond that of making persuasion in the souls of those who hear it (πειθὼ τοῖς ἀκούουσιν ἐν τῇ ψυχῇ ποιεῖν), to which Gorgias answers that it has none at all (453a).[31] Recalling the *Theaetetus* once again, we can see that although Gorgias' formulation of rhetoric here is different in certain ways from that of Protagoras on



sophistry, they nonetheless share a conception of exercising power through *logoi*, where the power operates *unidirectionally*, directed by those with wisdom or *technê* upon those without it—and, in Gorgias' mind, it divides those who are free from those who are enslaved through this daimonic work of persuasion.[32] Interesting, for our purposes, is the fact that Socrates says to Gorgias at this point that he is questioning him, in his words, "not for your sake, but for the sake of the *logos* [οὐ σοῦ ἕνεκα ἀλλὰ τοῦ λόγου], in order that it proceed in such a way that it make manifest to us, to the greatest extent, the issue we are discussing [ἵνα οὕτω προΐῃ ὡς μάλιστ᾽ ἂν ἡμῖν καταφανὲς ποιοῖ περὶ ὅτου λέγεται]" (453c). Socrates' distinction here between aiming his words at Gorgias in an *ad hominem* way and aiming them at the issue under discussion is surely ironic, since what is to be made clear by the *logos* is ambiguously both the power of rhetoric and the power of Gorgias as its practitioner. Rhetoric is the "artificer of persuasion" (πειθοῦς δημιουργός) (453a), and Gorgias himself claims to be its artificer (452d). Thus, in asking about the power of one, Socrates cannot avoid asking about the power of the other. Socrates has suspected that the persuasion Gorgias has in mind is the sort found in lawcourts and other public gatherings, dealing with what is just and unjust (ἅ ἐστι δίκαιά τε καὶ ἄδικα) (454b). Socrates urges Gorgias not to be surprised (θαυμάζῃς) by the questions he is asking him because they are for the sake of carrying through the *logos* in an orderly way (τοῦ ἐξῆς ἕνεκα περαίνεσθαι τὸν λόγον) (454b–c). He insists once again that he is not after Gorgias (οὐ σοῦ ἕνεκα) but is aiming to carry through what Gorgias is saying according to its underlying thought as he wishes (ἀλλὰ σὺ τὰ σαυτοῦ κατὰ τὴν ὑπόθεσιν ὅπως ἂν βούλῃ περαίνῃς) (454c).[33]

In the questioning that follows, Socrates has Gorgias agree that there is a difference between learning (μάθησις) and belief (πίστις) in that there may be false belief, but knowledge (ἐπιστήμη) is necessarily true (454d). Gorgias further agrees with Socrates that rhetoric is a manufacturer of persuasion leading to belief but not instruction in the matter of the just and the unjust (455a): "Therefore, the orator does not provide instruction [οὐδ᾽ ἄρα διδασκαλικὸς ὁ ῥήτωρ] to courts and other assemblies about things which are just and unjust. He merely persuades [ἀλλὰ πειστικὸς μόνον]. For after all, it would be impossible to instruct so large a crowd in a short time about matters of such importance" (455a). Socrates then raises the contrast between orators persuading people absent instruction and those authorities who have some specialized *technê* (455b–c)—a contrast underscored by Gorgias when he appeals to the power of Themistocles and Pericles, as opposed to the craftsmen, in counseling the production of public works (455e–456a). When Socrates acknowledges his own astonishment (θαυμάζων) at the fact that orators carry the day on such matters, and tells Gorgias that the power and greatness of rhetoric appears to him as something daemonic (δαιμονία γάρ τις ἔμοιγε καταφαίνεται τὸ μέγεθος οὕτω σκοποῦντι), Gorgias exclaims that rhetoric, so to speak, comprises in itself all powers (ἁπάσας τὰς δυνάμεις) (456a).[34] Gorgias argues further that the orator is able to "contest" (ἀγωνίζοιτο) any other expert practitioner in the public realm, and the orator is more persuasive before a multitude (πιθανώτερον … ἐν πλήθει) than a doctor or a member of any profession whatsoever (456c). Given that the power of this *technê* is so great and of this sort, rhetoric must be used like any other agonistic capacity (ἀγωνίᾳ). In other words, it must be used justly (δίκαιον)—its strength must not be misused (456c–457c). According to Gorgias, if a man becomes an orator and then uses this power unjustly, we should not hate the teacher who imparted the *technê*: it is the student who deserves to be expelled or put to death (457a–c).[35]

But the fact that Gorgias has raised, hypothetically, the possibility that an orator might fail to persuade others in the *polis* that he is using rhetoric justly reveals an inconsistency in his account that Socrates senses immediately. As we have seen in our treatment of the *Theaetetus*, the possibility countenanced by Gorgias here has as its parallel the possibility of a sophist failing to "change the appearances" on the part of his audience—which is the ability for which sophistry can claim to be a kind of wisdom worthy of recognition.[36] Before subjecting Gorgias to the inevitable *elenchos*, Socrates brings up the problem of dialogical partners misinterpreting disagreement as "being out to win" (φιλονικοῦντας) instead of inquiring into the matter put forward in *logos* (ἀλλ᾽ οὐ ζητοῦντας τὸ προκείμενον ἐν τῷ

λόγῳ) (457d). Telling Gorgias that he is afraid to refute him lest he be accused of being out to win (φιλονικοῦντα)—that is, "not speaking toward the issue and its becoming manifest, but toward you [οὐ πρὸς τὸ πρᾶγμα . . . λέγειν τοῦ καταφανὲς γενέσθαι, ἀλλὰ πρὸς σέ]"—Socrates says that to be subjected to *elenchos* is to undergo something beneficial, namely, "being delivered from the greatest of evils" (ἀπαλλαγῆναι κακοῦ τοῦ μεγίστου), for there is no evil so great for a human being as false opinion (δόξα ψευδὴς) about the matters in question (457e–458b).[37] Of course, the distinction posed again by Socrates in this passage—between speaking toward the issue under discussion and speaking toward Gorgias—is ironic: while Socrates appears to give Gorgias the freedom to opt out of the discussion, given the dramatic setting, Gorgias is not really free to do so, as Socrates is no doubt well aware. Indeed, when he tries to back out (458b–c) but is met with a clamor from the audience (458c–d), Gorgias makes the point explicit: "It would be shameful for me not to be willing, Socrates, after all this, especially when I profess to answer whatever anyone wishes to ask" (458d). Thus, the *elenchos* that Socrates has prepared for Gorgias is not simply a "refutation" of a claim but rather a showing up of Gorgias' sham "wisdom" to an audience of his admirers.[38]

Socrates begins the *elenchos* by focusing on what Gorgias has already conceded: namely, that rhetoric operates not through instruction but through persuasion (458e), which means that the orator is more persuasive to the ignorant (τοῖς μὴ εἰδόσιν) than to those who have knowledge (459a–b). Socrates puts it this way: "Rhetoric and the orator have no need to know how things really stand with things themselves [αὐτὰ . . . τὰ πράγματα]; they need only to discover some device of persuasion so as to appear to the unknowing to know more than those who know [μηχανὴν δέ τινα πειθοῦς ηὑρηκέναι ὥστε φαίνεσθαι τοῖς οὐκ εἰδόσι μᾶλλον εἰδέναι τῶν εἰδότων]" (459b-c). This raises the prospect of a divide between *ta pragmata* and Gorgianic discourse comparable to the divide we have identified between the *pragmateia* of dialogue and Protagoras' *homo mensura* doctrine. Socrates asks Gorgias whether the orator knows the very things themselves—the good and bad, beautiful and shameful, just and unjust—or whether he instead merely contrives to seem to know these matters (μεμηχανημένος ὥστε δοκεῖν εἰδέναι) (459d–e). Will the teacher of rhetoric merely make the pupil seem (δοκεῖν) to the multitude to know these things without really knowing them, and seem to be a good man without really being one? (459e). Gorgias has committed himself to revealing clearly the power of rhetoric, and yet Socrates' questions suggest that the power of rhetoric is not the power of knowing but a power premised on *doxa*, on the appearance of knowing. In other words, the power of rhetoric is premised on concealment, on *not* revealing and *not* making manifest what its power really is.[39] At this point, Gorgias faces a dilemma. His power and prestige as a teacher of rhetoric require that he stand by his promise to answer whatever he is asked—in this case, to reveal clearly the power of rhetoric; but to reveal this power is to be stripped of it, for the power itself depends upon concealing the fact that it does not communicate knowledge or instruction to the audience. Either route will bring about an *elenchos* of his account, and so, as Polus perceives a short time later, Gorgias is forced out of shame to contradict himself (461b).[40]

Gorgias responds that if his pupil does not know the things in question (good and bad, beautiful and shameful, just and unjust), then he will teach him these things (460a)—a point that Socrates seizes upon and has him repeat. From this, Socrates has Gorgias agree that a person who has learned what things are just is just, does just things, and will never wish to do injustice (460b–c). It follows, of course, that the orator is necessarily just and will never wish to do injustice (460c). But this contradicts Gorgias' account of how a pupil may use rhetoric unjustly, even though the teacher imparts it to be used justly. If the pupil can use rhetoric unjustly, it is not knowledge about the just and unjust. If rhetoric is knowledge about the just and unjust, then it is impossible for an orator to use it unjustly. But why does Gorgias agree with Socrates that the orator must know the just and the unjust? Why does he not take Polus' route and claim that the orator need not know these things about which he persuades the crowd? The answer is that Gorgias understands (as Polus does not) that the power of rhetoric depends on its reputation as knowledge—that its ability

to be more persuasive than the one who knows requires that the orator *seem* to have this knowledge.[41] Certainly, there are those "in the know" as to the fact that the orator need not have knowledge about those matters on which he persuades others, but nonetheless the orator is able to persuade people on these matters only by appearing to possess knowledge. Gorgias opts for self-contradiction in order to conceal that which is essential to the power of his *technê*. Thus, the *epideixis* Gorgias offers is an exhibition of self-concealment, precisely the recoiling from candidness (παρρησία) that is essential to rhetoric's power.[42] Consider, further, that Gorgias' retreat from candidness is dramatized by his allowing his pupil and inferior, Polus, to step in for him. Ironically, by allowing this to happen, Gorgias allows himself to be performatively contradicted, for his pupil demonstrates both that an orator need not have the knowledge that Gorgias claims to impart and that an orator does not, in fact, exercise the power over an audience that Gorgias proclaims.

Both Polus and Callicles will assert that the contradictions in Gorgias' account stem only from the fact that he was led, out of shame, to speak insincerely. Indeed, the fact that Socrates summarizes their conclusions to this point *as if Gorgias had been speaking sincerely* is what leads to Polus' exasperated outburst (461b–c). But that is just the point: the fact that Gorgias was driven into insincerity out of shame will serve as the basis for Socrates' account of rhetoric as a matter of pandering to the audience. According to Socrates, rhetoric is no *technê* at all but is only a "knack" (ἐμπειρία) for producing a kind of gratification and pleasure (χάριτός τινος καὶ ἡδονῆς ἀπεργασίας) (462b–c). With evident insincerity, Socrates tells Polus that he hesitates to say precisely what practice (ἐπιτήδευμα) rhetoric is a part of because he fears it may be even more ill-mannered (ἀγροικότερον) to state the truth, and he does not want Gorgias to think he is ridiculing his life's pursuit (διακωμῳδεῖν τὸ ἑαυτοῦ ἐπιτήδευμα) (462e).[43] When, in an absurd twist of irony, Gorgias responds by urging him not to be ashamed on his account, but to tell them what it is, Socrates replies that rhetoric is a practice (ἐπιτήδευμα) "belonging to a soul given to boldness, shrewd at guesswork, naturally clever in intercourse with people [ψυχῆς δὲ στοχαστικῆς καὶ ἀνδρείας καὶ φύσει δεινῆς προσομιλεῖν τοῖς ἀνθρώποις]" (463a–b).[44] Clearly, this is aimed at Gorgias, who is said to turn his students into clever speakers (ἀλλὰ λέγειν οἴεται δεῖν ποιεῖν δεινούς) (*Meno* 95c). But more to the point, the "boldness" and "shrewdness" in Socrates' description is evidenced in Gorgias' performance thus far, since he is willing to contradict himself to spare his reputation and shrewdly guessing at what the crowd wants to hear. Yet, in feeling shame before the audience and altering his account accordingly, Gorgias performatively contradicts his account of the power of rhetoric: rather than enslaving his audience, the orator panders to them, subordinating himself to their expectations.[45] Socrates' prepared speech on rhetoric as pandering thus restates what has happened dramatically in the dialogue. It is with an eye to Gorgias' insincerity that Socrates foregrounds the *doxa* and self-concealment on which the so-called "power" of rhetoric depends.[46]

Socrates continues with his response that the sum and substance of rhetoric is pandering (κολακεία), a life-structuring practice that has many other parts, including cookery, cosmetics and sophistry (463b).[47] To clarify his account of rhetoric, Socrates gets Gorgias to agree with him that there is something called body and something called soul (463e), that there is a healthy condition (εὐεξία) of each, and further that there is a condition that seems (δοκοῦσαν) healthy without being so (οὖσαν δ᾽ οὔ) (464a); correlatively, there are *technai* serving the body, gymnastics and medicine, and *technai* serving the soul, law-giving (νομοθετική) and justice (δικαιοσύνη)—the latter together comprising the political (464b). While these four *technai* always serve what is best (πρὸς τὸ βέλτιστον θεραπευουσῶν) for the body and soul, respectively, pandering cares nothing for what is best but instead puts on the mask of each, pretending to be the character she puts on (464c-d).[48] Socrates then gets to the point of his speech on rhetoric: "I call it pandering and I say the thing is ugly and shameful—this I direct to you, Polus—because it shrewdly guesses at what is pleasant, omitting what is best. And it is no *technê*, I claim, but only a knack [*empeiria*], for it has no *logos* of what it administers or of those to whom it administers, with the result that it



cannot state the cause of each treatment. I do not give the name of *technê* to a thing that is without *logos* [ἄλογον πρᾶγμα]" (465a). In this passage, Socrates says explicitly with respect to Polus what he repeatedly denied with respect to Gorgias, namely, that the *logos* is directed at him (τοῦτο γὰρ πρὸς σὲ λέγω). Of course, Socrates' *logos* is aimed not only at Polus but at Gorgias as well, the teacher of rhetoric, for Socrates' words here recall Gorgias' inability to provide a *logos*, his defeated attempt to reveal the power of his rhetoric.

Socrates' *elenchos* of Gorgias has carried out a radical inversion of Gorgias' position: *logos*, as wielded by a Gorgianic orator, turns out to be a matter of pandering to the audience rather than enslaving them through the irresistible power of persuasion. This radical inversion of Gorgias' doctrine of *logos* has occurred by way of a performative contradiction where it is precisely Gorgias' *own performance* in how he (insincerely) answers Socrates' questions that leads to the inversion. As such, the performative contradiction dramatized in the *Gorgias* is comedic in the same way that the performative contradiction involving Protagoras' *homo mensura* doctrine in the *Theaetetus* is comedic. There are, to be sure, Gorgianic grounds for Plato to dramatize such comedies of performative contradiction— following Gorgias' recommendation (preserved in Aristotle's *Rhetoric*) to destroy (*diaphtheirein*) an opponent's seriousness with laughter and an opponent's laughter with seriousness (1419b3–5). But beyond this, there is evidence in the dialogue that Gorgias' acolytes, Polus and Callicles, recognize (even if they do not appreciate) the comedic aspect of what Gorgias endures at the hands of Socrates, from Polus' exasperated response to Socrates' *elenchos* of Gorgias (461b–c), to Callicles' question as to whether Socrates is serious or just playing with them (481b), to the multiple instances of Polus and Callicles laughing at Socrates outright or ridiculing his philosophical practice (e.g., 473e, 482c, 485a–e). Indeed, there are moments of performative contradiction in Socrates' exchanges with Polus and Callicles in the later parts of Plato's *Gorgias*, and these episodes are comedic for the reasons we see in the episodes involving Gorgias and Protagoras, as discussed above.[49]

## 4. Conclusions: What Performative Contradiction Reveals

So what, then, should we take away from these parallel dramatizations of performative contradiction—of Protagoras' *homo mensura* doctrine in Plato's *Theaetetus* and of Gorgias' account of the power of *logos* in Plato's *Gorgias*? Three points, in particular, seem to be of chief philosophical significance.

The first point relates to the close proximity between Socrates' elenctic practice and sophistry. Socratic *elenchos* operates by way of a theater of appearing—tracked throughout by the language of *dokein/phainesthai*—which reveals it to be something very much like the "making-appear" and "changing the appearances" theorized by Protagoras in the *Theaetetus*. Indeed, Gorgias, himself, in his *Encomium of Helen*, brings up the *logoi* of philosophers and astronomers as a clear attestation of the ability to change how things appear in the sense of replacing one *doxa* with another: astronomers "make what is incredible and unclear seem apparent to the eyes of opinion [τὰ ἄπιστα καὶ ἄδηλα φαίνεσθαι τοῖς τῆς δόξης ὄμμασιν ἐποίησαν]", while in philosophical *logoi* "swiftness of thought is also shown making belief in an opinion easily changed [δείκνυται καὶ γνώμης τάχος καὶ ... ὡς εὐμετάβολον ποιοῦσι τὴν τῆς δόξης πίστιν]" (13).[50] The phainomenal character of Socratic *elenchos* is presented in a passage from Plato's *Sophist* that is illuminating for our purposes. There, the Eleatic Stranger juxtaposes *elenchos* with the time-honored method of scolding or admonishing (νουθετητική) as an alternative way to "get rid of the belief in one's own wisdom" (230a–b), and says the following about those who practice *elenchos*:

> They cross-examine someone when he thinks he's saying something though he's saying nothing [διερωτῶσιν ὧν ἂν οἴηταί τίς τι πέρι λέγειν λέγων μηδέν] ... They collect his opinions together during the discussion, put them side by side [καὶ συνάγοντες δὴ τοῖς λόγοις εἰς ταὐτὸν τιθέασι παρ᾽ ἀλλήλας], and show [ἐπιδεικνύουσιν] that they conflict with each other at the same time on the same subjects in relation to the same things and in the same respects. The people who are being examined see this, get angry at themselves, and become calmer

toward others [οἱ δ᾽ ὁρῶντες ἑαυτοῖς μὲν χαλεπαίνουσι, πρὸς δὲ τοὺς ἄλλους ἡμεροῦνται]. They are set free from their inflated and rigid beliefs about themselves that way [καὶ τούτῳ δὴ τῷ τρόπῳ τῶν περὶ αὐτοὺς μεγάλων καὶ σκληρῶν δοξῶν ἀπαλλάττονται], and no setting-free [ἀπαλλαγῶν] is more pleasant to hear or has a more lasting effect on them … The people who purify the soul, my young friend, likewise think that the soul won't get any advantage from any learning that's offered to it until the one doing *elenchos* puts the one undergoing *elenchos* into a state of shame, [πρὶν ἂν ἐλέγχων τις τὸν ἐλεγχόμενον εἰς αἰσχύνην καταστήσας] removes the opinions that interferes with learning, and shows it forth purified, believing that it knows only those things that it does know, and nothing more [τὰς τοῖς μαθήμασιν ἐμποδίους δόξας ἐξελών, καθαρὸν ἀποφήνῃ καὶ ταῦτα ἡγούμενον ἅπερ οἶδεν εἰδέναι μόνα, πλείω δὲ μή]. (*Sophist* 230b–c)

So the Eleatic Stranger concludes: "For all these reasons, Theaetetus, we have to say that *elenchos* is the authoritative and most important kind of purification [μεγίστη καὶ κυριωτάτη τῶν καθάρσεών ἐστι]" (230d), and he calls this activity by which "empty belief in one's own wisdom" is refuted (ὁ περὶ τὴν μάταιον δοξοσοφίαν γιγνόμενος ἔλεγχος) a kind of sophistry—namely, "sophistry of noble lineage [ἡ γένει γενναία σοφιστική]" (231b).[51]

For our purposes, the katharatic *elenchos* described by the Eleatic Stranger would appear to be a highly idealized portrait of Socratic practice; it is dubious that we find anything like these results in Plato's *Gorgias*. Nonetheless, in this passage from the *Sophist*, as in the *Theaetetus* and *Gorgias*, it is the epideictic character of *elenchos*, its capacity to show or exhibit something "to the eyes of opinion", as Gorgias might say, that reveals how close it is to the sophistic power theorized by Protagoras and described by Gorgias in his *Encomium of Helen*. Socratic *elenchos*, like the *logoi* wielded by Protagoras and Gorgias, aims at effecting a change in the interlocutor or audience—and we should note that it is not only the language of *epideixis* that we find in this passage from the *Sophist* but also the language of *apallangê*, a "setting-free" or "being-released" from that relation to oneself that prevailed before the exercise of *logos* (cf. *Theatetus* 168a–b).[52] According to the Eleatic Stranger, "no setting-free [*apallangê*] is more pleasant to hear or has a more lasting effect on them [πασῶν τε ἀπαλλαγῶν ἀκούειν τε ἡδίστην καὶ τῷ πάσχοντι βεβαιότατα γιγνομένην]" than *elenchos* (230b).

The second take-away, I would argue, is that we find an illuminating parallel between the self-reflexivity at work in the performative contradictions of these sophists and the self-reflexivity in Plato's own dialogue-form of writing.[53] There is, of course, self-reflexivity in Protagorean thought—and indeed, as Burnyeat has suggested, it may be that the two *logoi* fragment of Protagoras (DK 80B6a; compare 80A1) reflects the *homo mensura* doctrine applied to itself, self-reflexively—but Gorgias' writings are an even more fitting parallel for Plato's dialogues with respect to self-reflexivity.[54] For example, in the *Encomium of Helen* we find self-reflexivity in multiple forms, from Gorgias avowedly making his case to the *doxa* of the audience (8–9), to his theorizing the power of *logos* to replace one *doxa* for another (δόξαν ἀντὶ δόξης τὴν μὲν ἀφελόμενοι τὴν δ ἐνεργασάμενοι) (13), to his referring to his written work at the end as a *paignion* (21). These complex levels of attention-to-oneself reflected in Gorgias' writing are of a piece with Plato dramatizing the forgetfulness-of-oneself that emerges through performative contradiction.[55] These parallels between Plato the writer and the sophistic theorists perhaps work against the tendency—so strong in the scholarly literature—to draw sharp boundaries between Plato and the sophistic thinkers. For scholars as different in their commitments and interpretative perspectives as Terence Irwin, Jacqueline De Romilly and Scott Consigny, Plato's Socrates is a thinker aiming at philosophical objectivity, foundationalism, and characterless rationality, and in this way represents a radical departure from the sophists' human-oriented practical concerns, relativism, and commitment to something akin to "anti-foundationalism."[56]

Ironically, however, Plato's recurrent portrayal of Socrates subjecting sophistic theories to performative contradiction suggests something very different. The problem with sophistic theories like Protagoras' *homo mensura* and Gorgias' doctrine of *logos* is that they forget the human-all-too-human conditions for their own possibility. Indeed, it is their detachment from the realm of human interaction, where the real-life concerns of danger, sickness, and death give rise to the urgent need for wisdom, and where *logos* is exercised reciprocally rather than unidirectionally, that is "demonstrated" through the performative contradiction dramatized in *Gorgias* and *Theaetetus*.

Third and finally, then, performative contradiction as dramatized in the case of sophistic theories has the effect of focusing our attention on the situatedness-within-the-human-world that constitutes the shared condition for the possibility of sophistry and Socratic *dialegesthai*. In the *Theaetetus*, Socrates was in the position of reminding Protagoras of what is forgotten when the *homo mensura* doctrine is put forward in dialogue: namely, the whole *pragmateia* of *dialegesthai* and Socratic midwifery, which requires a venue for assessing and critiquing one another's opinions. In the *Gorgias*, Socrates directs our attention to the human audience for Gorgianic rhetoric, which does not represent the vulnerable patient to whom the orator unidirectionally directs his *logoi*, analogous to a physician administering drugs. Rather, performatively, this audience represents the crowd the orator panders to and shrewdly gratifies as he tailors his words. The reciprocal, rather than unidirectional, relations of power within the *pragmateia* underlying rhetorical discourse are what Socrates uncovers through *elenchos*. Indeed, as the dialogue advances with Polus and then Callicles as interlocutors, Socrates will direct them toward recognizing how dialectical refutation relies on the kind of engagement by the participants that is expressive of the character-traits of knowledge (ἐπιστήμη), goodwill (εὔνοια) and outspokenness (παρρησία) (487a). Accordingly, what emerges in the dialogue as a contrast to Gorgianic rhetoric is a dialogical format that thrives on an openness to the mutual critique of opinions.[57] Whereas Protagorean *logos* involves an epistemic equalizing of participants to the point of undifferentiation, Socratic *elenchos* singles out participants, focusing critical attention on whether one is in harmony with oneself and whether one's position in dialogue holds its integrity in the face of questioning. By vivid contrast with the *homo mensura* doctrine, Socrates' practice of *elenctic dialegesthai* presupposes that one may be wrong in one's beliefs, even wrong about what one believes, and that one's power is measured by *logos*, not something one exercises by "using" *logos* as an instrument or weapon. It is these differences between Socrates and his sophistic rivals that are brought to light through the performative contradictions dramatized in Plato's texts.[58]

**Funding:** This research received no external funding.

**Institutional Review Board Statement:** Not applicable.

**Informed Consent Statement:** Not applicable.

**Data Availability Statement:** Not applicable.

**Conflicts of Interest:** The authors declare no conflict of interest.

## Notes

[1]    Compare Sextus Empiricus, *Against the Mathematicians* VII.60 (=DK 80B1) and its formulation in Plato's *Theaetetus* at 151e. In what follows, I will also discuss the parallel refutation of Protagoras' *homo mensura* doctrine in Plato's *Cratylus* 385e–387a, though in a much less detailed way than what is found in the *Gorgias* and *Theaetetus*. See also *Euthydemus* 286a-287a for relevant attention to the problematic character of the *homo mensura* doctrine.

[2]    For a persuasive account of Protagoras' commitment to humanism, if not relativism, see (Versenyi 1962). For an interpretation of Protagoras generally that emphasizes relativism, see (De Romilly 2002). For an argument expressing doubts about Protagoras' relativism, and doubts about sophistic theories being generally "relativistic", see (Bett 1989). For discussion of how later Greek philosophy interpreted Protagoras' *homo mensura* doctrine as subjectivist so as to entail the idea (also attributed to Protagoras) that "it is possible to dispute with equal validity on either side of every question, including the question whether it is possible to dispute with equal validity on either side of every question", see (Burnyeat 1976a, pp. 60–61). For the argument that there is a

robust anti-relativism presented in Plato's portrait of Protagoras in *Protagoras*, see (Taylor and Lee 2020); for a contrary view, see (Sentesy 2020).

3    On the question of whether Gorgias is a sophist, Taylor and Lee (2020) note the following: "At *Apology* 19e–20c Plato represents Socrates as naming four individuals who undertake to teach or educate people (*paideuein anthrōpous*) in return for fees; they are Gorgias (from Leontini in Sicily), Hippias (from Elis, in the north-western Peloponnese), Prodicus (from Ceos, off the southern tip of Attica) and Euenus (from Paros, in the southern Aegean). Of the four only Euenus is expressly said to teach "human and political excellence" (*tēs . . . arêtes . . . anthrōpinēs te kai politikēs*, i.e., success in the running of one's life and in public affairs), but the context strongly suggests that the other three are seen as offering the same kind of instruction." Schiappa (1999) warns against the overgeneralization implicit in using the term "sophistic", especially with respect to Gorgias, 56.

4    I have noted that 'Protagoras' is ventriloquized by Socrates in the *Theaetetus* in order to mark the asymmetry between Gorgias and Protagoras in these two dialogues: Gorgias as a character articulating his own views in dialogue with the character of Socrates in Plato's *Gorgias*, and 'Protagoras' as a construction ventriloquized by the character Socrates in dialogue with other interlocutors in the *Theaetetus*. See Barney (2006) for the argument that "Gorgias and Protagoras can plausibly be seen as forming a united front of deflationary anti-realism . . . There is no reality beyond appearance, and no hope for any knowledge which would be different in kind from our fallible opinions" (p. 94). Consider, further, that Plato's Socrates treats rhetoric and sophistry as proximate (*engus*) practices that are at issue in Socrates' refutation of Gorgias (at *Gorgias* 465c, *passim*). See the discussion of these issues in (Tusi 2020).

5    Certainly some of Plato's dialogues are more robustly dramatic and "dialogical" than others, and some are more didactic and "monological" in structure. I have addressed this phenomenon across Plato's texts in (Metcalf 2004, 2006, 2015, 2018).

6    For example, the treatment of Protagoras' *peritrope* or "self-refutation" in (Passmore 1961; Mackie 1964; Burnyeat 1976a, 1976b; Bett 1989; Chappell 2005, 2006).

7    Yet, the *peritrope* argument as applied to Protagoras is criticized in Chappell (2006): "Protagoras never claims that 'Every appearance is true'; he claims only that 'Every appearance is true for the person to whom it appears.' . . . The *Peritrope* does not disprove this thesis; indeed it does not even address it. There is no inconsistency between 'It is true for Protagoras that every appearance is true for the person to whom it appears' and 'It is true for someone else that not every appearance is true for the person to whom it appears.' There isn't even an inconsistency between 'It is true for Protagoras that every appearance is true for the person to whom it appears' and 'It is true for someone else that not every appearance is true period or *simpliciter* (absolutely true).' Sextus' argument works against the claim that every appearance is non-relatively true; but Protagoras' claim is only that every appearance is relatively true. So if Sextus' argument is understood in the most obvious way, as aiming to refute Protagoras by showing that he contradicts himself, it misses its target" (p. 110).

8    (Burnyeat 1976a, p. 48). Burnyeat expands on his explanation when he goes on to write, "Add to this evidence the frequency of phrases like *peritrepein heauton*, to refute oneself (*PH* 1.122, 2.188; *M* 8.331a, 360, 463, 10.18), and the interpretation of *peritrope* as self-refutation becomes compelling. For precisely what self-refutation consists in is a reversal whereby advancing a proposal commits one to its contradictory" (p. 49).

9    See (Passmore 1961): "So if Protagoras is correct, it will follow that man is the measure of all things (since this is how it appears to Protagoras) and that man is *not* the measure of all things (since this is how it appears to his opponents). Hence his theory is in a precise sense self-contradictory" (p. 67). Burnyeat (1990) concludes in a similar fashion to Passmore (1961): "Isn't there something inherently paradoxical about someone asserting (or believing) that *all truth is relative*? That proposition sums up the message of a completely general relativism, but when asserted it is propounded as itself a truth. The reason for this is simple but fundamental: to assert anything is to assert it as a truth, as something which is the case . . . Relativism is self-refuting, and for reasons that go deep into the nature of assertion and belief" (p. 30), here citing a parallel argument in Edmund Husserl's *Logical Investigations*, 2nd edition (Husserl 1970), 139. Again, Aristotle's *Metaphysics*, book *Gamma*, articulates this point: "Moreover it follows that all statements would be true and all false [πρὸς δὲ τούτῳ ὅτι πάντες ἂν ἀληθεύοιεν καὶ πάντες ἂν ψεύδοιντο]; and that our opponent himself admits that what he says is false [καὶ αὐτὸς αὑτὸν ὁμολογεῖ ψεύδεσθαι]. Besides, it is obvious that discussion with him is pointless [ἅμα δὲ φανερὸν ὅτι περὶ οὐθενός ἐστι πρὸς τοῦτον ἡ σκέψις], because he makes no real statement [οὐθὲν γὰρ λέγει]. For he says neither 'yes' nor 'no,' but 'yes and no'; and again he denies both of these and says 'neither yes nor no'; otherwise there would be already some definite statement" (1008a28–32). On the relations between Aristotle's *Metaphysics*, book Gamma, and Protagorean relativism, and Plato's Theaetetus, see the discussion in (Long 2006), pp. 49–60.

10   Passmore (1961) uses the example of speaking the words, "I cannot speak", to illustrate a pragmatic self-refutation (p. 80). Mackie (1964) writes: "In pragmatic self-refutation the way in which an item happens to be presented conflicts with the item itself. But where we find operational self-refutation there is no other way in which this precise item can be presented" (p. 197). For example, Mackie argues that "I am not thinking right now" is operationally self-refuting, not a case of merely pragmatic self-refutation (p. 198). Following Mackie (1964), Burnyeat (1976a) uses the example of writing that I am not writing: "If I whisper that I am not writing, what I say may well be true, but if I write it, it must be false" (p. 52). Implicit in this example is the sense, expressed by numerous commentators, that pragmatic self-refutations are not as decisive, philosophically, as the "operational" or "absolute" self-refutation captured in the *peritrope*, since the one who has been pragmatically self-refuted may change the manner by which they put forward the thesis, thereby evading self-refutation.

11    (Burnyeat 1976a, p. 59). See also (Burnyeat 1976b, p. 172): "It is this dialectical setting which provides the key to Protagoras' self-refutation" (p. 172). Aristotle seems to have this in mind in his account of the self-refuting character of denying the principle of non-contradiction, in *Metaphysics* (book *Gamma*), for he remarks that the "the person responsible" for this *elenctic* demonstration of the principle is "not he who demonstrates but he who acquiesces [ἀλλ᾽ αἴτιος οὐχ ὁ ἀποδεικνὺς ἀλλ᾽ ὁ ὑπομένων]; for though he disowns reason he acquiesces to reason [ἀναιρῶν γὰρ λόγον ὑπομένει λόγον]" (1006b13ff.). Similarly, Chappell (2005) writes: "The deepest difficulty with a Protagorean relativist is not to refute his argument [but] to see what he says as an *argument* at all" (p. 114).

12    For the connection between the dramatic framing/narrative of the *Theaetetus* and Protagorean relativism, see (Schultz 2020, pp. 21–23).

13    Compare Socrates' remarks at the end of the dialogue: "Well now, dear lad, are we still pregnant, still in labor with any thoughts about knowledge? Or have we been delivered of them all? . . . And so, Theaetetus, if ever in the future you should attempt to conceive or should succeed in conceiving other theories, they will be better ones as a result of this enquiry [διὰ τὴν νῦν ἐξέτασιν]. And if you remain barren, your companions will find you gentler and less tiresome; you will be sound-minded [σωφρόνως] and not think you know what you don't know [οὐκ οἰόμενος εἰδέναι ἃ μὴ οἶσθα]. This is all my art can achieve—nothing more. I do not know any of the things that other men know [οὐδέ τι οἶδα ὧν οἱ ἄλλοι]—the great and inspired men of today and yesterday [ὅσοι μεγάλοι καὶ θαυμάσιοι ἄνδρες εἰσί τε καὶ γεγόνασιν]. But this art of midwifery my mother and I had allotted to us by God; she to deliver women, I to deliver men that are young and generous of spirit, all that have any beauty [τῶν νέων τε καὶ γενναίων καὶ ὅσοι καλοί]" (*Theaetetus* 210b–d).

14    Sentesy (2020) discusses at length the claim that for Protagoras there is no difference between being and appearing, at issue in this passage.

15    But see (Versenyi 1962; De Romilly 2002) on the essentially *human* character of Protagorean relativism. De Romilly (2002) writes: "[The Sophists] were the first to try to think of the world and life purely in terms of human beings. They were the first to consider the relativity of knowledge as a fundamental principle, and to open up the way not only for free-thinking but also for absolute doubt regarding all metaphysical, religious, and moral matters . . . In this world of theirs, the necessities of communal life created a new place and a new meaning for justice, concord, and the human virtues in general. All humanist systems of thought which create values within an existentialist framework sprang from the seeds sown by the Sophists' new ideas" (p. 238).

16    Notice that Plato's *Euthydemus* also addresses the performative contradiction at issue in the *homo mensura* doctrine when Euthydemus and Dionysodorus posit the impossibility of speaking falsely—and Socrates remarks on how this doctrine is self-overturning (286c)—for the very same reasons that we see in *Theaetetus*. Most relevant is the fact that, according to Socrates, it would rule out the very possibility of *elenchos* since it precludes ever having false beliefs or 'being wrong' (286d-e). On this point, see Long (2004): "Relativism leaves refutation both pointless and invalid. If we are all infallible Measures, what could be gained from comparing opinions or testing one another's views? How, furthermore, could we ever find weaknesses in other people's set of convictions? . . . I suggest that Plato's intention here is to see refutation as posing a particular dilemma for the relativist" (p. 25).

17    Here, I am bringing together the *pragmateia* in light of which the *homo mensura* doctrine is performatively contradicted with the use of *pragmata echein* at 174b. LSJ cites examples of *pragmata* or *pragmata echein* used in the sense of "going to the trouble" or "exerting oneself" in Plato's texts at *Apology* 41d, *Phaedo* 115a, and *Republic* 406e.

18    Long (2004) notes that at *Protagoras* 336b Socrates distinguishes between *dialegesthai* and *dêmêgorein*, and Socrates calls his own arguments against Protagoras *dêmêgorein* at *Theaetetus* 162d.

19    On this point, see (Bell 2011): "'The many,' especially when they are gathered together—as in the assembly—are not many, but one; and this, paradoxically, is no less the case when—as in the assembly—dissent, difference and disagreement are expressed" (p. 385). In a footnote, Bell continues his analysis of the "remarkable proximity" between Plato's analysis and Heidegger on *Dasein*'s everydayness: "To say that the many are a singularity is to say that everyone is (the same as) the other—one is the others and the others are one: one thinks what the many think, is pleased and pained by what pleases and pains the many, is moved and persuaded by what moves and persuades the many and speaks what the many speak . . . Dogma names that way of being in which one is the others with whom one exists. It names, therefore, the way of being in which one is distanced from and has forgotten oneself" (392f16).

20    Admittedly, this account of demagogic rhetoric in Plato's *Republic* is not one that merely applies to democratic cultures. Konrad Heiden offers an account very much like that in the *Republic* when, in his book on the rise of Hitler in the 20th century, he says the following of the orator/propagandist: "Like a piece of wood floating on the waves, he follows the shifting currents of public opinion. This is his true strength . . . The speaker is in constant communication with the masses; he hears an echo and senses the inner vibration . . . When a resonance issues from the depths of the substance, the masses have given him the pitch; he knows in what terms he must finally address them . . . This mass, with its anonymous intellectual pressure, its unexpected moods and unconscious desires, mirrors and echoes the commanding force of prevailing conditions . . . It is the art of the great propagandist to detect this murmur and translate it into intelligible utterance and convincing action. If he can do this, his utterances and actions may be full of contradictions—because the contradictions lie in the things themselves" (Heiden 1968, pp. 140–41).

21    Concurring with McDowell (1973) that Socrates is "charitably altering the original position on Protagoras' behalf, so as to render it (as he sees it) more defensible. Cf. (Bett 1989, p. 166 n.58).



22    I believe that this parallel between Protagoras in the *Theaetetus* and Gorgias in the *Gorgias* is missed in Bett (1989), where it is argued that a focus on persuasion need not entail relativism, which is true enough. However, the relativism articulated by Socrates/Protagoras in the *Theaetetus* is a relativism relative-to-seeming/appearing, and "sophistry" as theorized by Protagoras signifies the essentially *rhetorical* power of using *logoi* to make appear or change the appearances.

23    In the *Encomium of Helen*, Gorgias likens the way that the soul is affected by *logos* to the way that the body is affected by a *pharmakon*: "The power [δύναμις] of *logos* has the same relation to the order of the soul as the order of *pharmaka* has to the nature of bodies. For just as different *pharmaka* expel different humours from the body, and some end illness while others end life, so some *logoi* induce pain, some pleasure, some fear, some courage in those who hear, while others drug and bewitch the soul with a kind of bad persuasion [πειθοῖ τινι κακῇ]" (§14). See Drake (2021) on the significance of "bodies" in Gorgias' text.

24    As to the comedic character of this refutation, Ewegen (2014) writes: "One could say that the *Cratylus* is the full and comic articulation of the second consequence of Protagoras' doctrine that 'the human being is the measure of all things' as it is analyzed by Socrates in the *Theaetetus*. The *Cratylus* presents, in vivid and comic detail, the devastating dissolution of *logos* that follows upon the Protagorean position" (p. 69).

25    See a plausible hypothesis as to the battle in which Theaetetus was mortally wounded in (Nails 2002, pp. 276–77).

26    The Greek word *epideixis*, in both its substantive and verbal forms, is used five times in the first Stephanus page of the dialogue, underscoring its dramatic significance for what follows.

27    Nichols (1998) notes: "One can hardly doubt that Socrates already knew Gorgias to be a rhetorician. Furthermore, it becomes altogether clear early in Socrates' discussion with Polus that Socrates has quite a fully developed conception of what something called rhetoric is . . . " (131). On Socrates' use of rhetoric in the *Gorgias* more generally, see (Roochnik 1995).

28    To put it a bit differently, the *elenchos* of Gorgianic rhetoric dramatized in Plato's *Gorgias* operates quite differently from Socrates' own account of *elenchos* in the *Apology*, where he portrays his going from one person with a reputation for wisdom to another such person as though, in each case, he had an open mind as to whether such a person might actually have the wisdom they are reputed to have (*Apology* 21e–22a). Indeed, Socrates' account in the *Apology* suggests that he goes into these elenctic examinations on the assumption that those with reputations for wisdom do have the wisdom that they are reputed to have, for he claims to have begun these examinations only to refute the oracular proclamation that "No one is wiser than Socrates" (21a–b). What is more, he portrays himself as having been surprised to discover, through elenctic examination, that the various reputations for wisdom within the *polis* are either entirely empty or else they mislead those who enjoy these reputations into thinking that they have wisdom other than the limited sort that they do have (see *Apology* 22d–e). On the question of whether Socrates goes into the encounter with an open mind as to whether Gorgias indeed has the power and *technê* that he is reputed to have, see (Ewegen 2020) for criticism of my interpretation of Socratic irony, and my response in (Metcalf 2022).

29    See (Consigny 2001) for a reasoned objection to this very distinction as applied to Gorgianic rhetoric.

30    Dodds (1959, p. 202) notes that the expression, "freedom and/or rule over others [*eleutherias ē allōn archēs*]", appears in Thucydides' account of the Mytilean debate in *The Peloponnesian War* (3.45.6).

31    Here, Socrates introduces for the first time in the dialogue the word, "soul" (ψυχή) which is central to Gorgias' portrait of the orator's power in *Encomium of Helen*, and which will become increasingly important in Plato's *Gorgias*. Segal (1962) makes the point this way: "The *techne* of Gorgias rests upon a 'psychological' foundation: it is at least assumed that the psyche has an independent life and area of activity of which the rhetor must learn and which to some extent he must be able to control" (pp. 105–6).

32    The parallel account of rhetoric's unidirectional exercise of power in Gorgias' texts is in the *Encomium of Helen*: "For the *logos* which persuades the soul constrains [ἠνάγκασε] the soul which it persuades, both to obey its utterances [πιθέσθαι τοῖς λεγομένοις] and to approve its doings [καὶ συναινέσαι τοῖς ποιουμένοις]" (§12, in part). It is also found, of course, in the famous lines in §8: "*Logos* is a great master [δυνάστης μέγας], which accomplishes divine deeds with the smallest and least apparent of bodies; for it is able [δύναται ] to stop fear, remove pain, implant joy and augment pity."

33    On the ironic distinction between speaking *pros to pragma* and speaking *pros tina*, consider the passage in Plato's *Laches* in which Nicias says to Lysimachus: "You seem to me not to know that whoever comes into close contact with Socrates and associates with him in conversation must necessarily, even if he began by conversing about something else at first, keep on being led around by the man's *logoi* until he submits to giving an account of himself [τὸ διδόναι περὶ αὑτοῦ λόγον]—concerning both his present manner of life and the life he has lived hitherto [ὄντινα τρόπον νῦν τε ζῇ καὶ ὄντινα τὸν παρεληλυθότα βίον βεβίωκεν]. And when he does submit to this questioning, you don't realize that Socrates will not let him go before he has well and truly tested every last detail [βασανίσῃ ταῦτα εὖ τε καὶ καλῶς ἅπαντα] . . . For me there is nothing unusual or unpleasant in being tested [βασανίζεσθαι] by Socrates, but I realized some time ago that, if Socrates were present, the *logos* would not be about the boys but about ourselves [ἀλλὰ περὶ ἡμῶν αὐτῶν]" (187e–188c).

34    Notice that this exclamation follows Gorgias' remark, "Well, I will try, Socrates, to reveal to you clearly the whole power of rhetoric [σαφῶς ἀποκαλύψαι τὴν τῆς ῥητορικῆς δύναμιν ἅπασαν]" (455d). Later Socrates riffs on Gorgias' remark when he says: "I beg you in the name of Zeus, reveal [ἀποκαλύψας] what the power of rhetoric is, as you promised" (459e–460a).

35    Here, Gorgias articulates the character of rhetoric as an *agônia* when he tells Socrates that "the orator is able to speak against everyone and on every question in such a way as to win over the votes of the multitude, practically in any matter he may

choose to take up" (457a–b)—a remark that echoes distinctly his standing claim to be able to answer any question posed to him. Benardete (1991) identifies the "inconsistency" for which Gorgias will suffer *elenchos* as follows: if the rhetorician has the power Gorgias says he has, it is impossible that the unjust rhetorician could ever be found to be unjust (p. 24). Put differently, if the rhetor gets caught, either the city must know what the rhetor does not, or the rhetor does not have an art (p. 28).

36    As Dodds 1959 notes, the fundamental significance of *doxa* for *peithein* is made vividly clear in Plato's *Theaetetus*, in which Socrates has Theaetetus agree that to persuade (τὸ πεῖσαι) is to make opine (δοξάσαι ποιῆσαι) (201b).

37    Notice that Socrates' words here about being committed to *elenchos* will be repeated in a more compressed form at the end of his exchange with Gorgias (461a).

38    Gorgias' reputation is key to Socrates' remarks, for Socrates says that the *elenchos* which he is ready to spring aims to deliver Gorgias from the greatest evil (δόξα ψευδὴς) about the topics under discussion. *Doxa pseudês* does mean "false opinion", of course, but it also means "false reputation", and this ambiguity is crucial, as noted by Benardete (1991, p. 25).

39    On this point (Stauffer 2006) writes: "Perhaps if [Gorgias'] art were indeed all-powerful, he would have no need to worry about its public reputation. But the power of rhetoric is not so great that it can overcome the need for concealment" (p. 33).

40    Beversluis (2000) offers a different reading: Gorgias "carelessly assents to . . . theory-laden propositions heavily infused with contra-endoxic Socratic doctrine which he does not believe and which . . . Socrates knows he does not believe" (p. 311).

41    Benardete (1991) writes that Polus is willing to sacrifice the rationality of rhetoric if he can retain its power. On my reading, Polus fails to understand that rhetoric exercises power only if it *appears* to be rational, p. 33.

42    Consigny (2001) interprets Gorgias' "anti-foundationalist" stance to be more transparent than the stealth "foundationalism" of Plato's Socrates: "For Gorgias, the Socratic strategy of self-effacement is the clever pose of the person who wishes to conceal his foundationalist commitment; and in this way it betrays deception rather than objectivity. Socrates' ironic profession of ignorance is of course a sham, for he *does* believe that he knows what is most basic to the foundationalist position, namely that there is an objective truth that antedates human inquiry" (pp. 193–94).

43    Indeed, in the very next sentence, Socrates casts a barb at Gorgias when he says, "Actually, I don't really know whether what I call rhetoric is the rhetoric Gorgias practices [ἐπιτηδεύει], for the *logos* did not make clear to me what he believes it is [καὶ γὰρ ἄρτι ἐκ τοῦ λόγου οὐδὲν ἡμῖν καταφανὲς ἐγένετο τί ποτε οὗτος ἡγεῖται]" (462e–463a).

44    What is fascinating about this statement is that it is Socrates who has thus far demonstrated boldness by challenging Gorgias at a party of orators, Socrates who has been shrewd at guessing the meaning of his interlocutors and anticipating their responses, Socrates who has proven himself most clever (*deinos*). Compare Plato's *Apology* 17b, in which Socrates explicitly denies being a "clever speaker."

45    Tarnopolsky (2010) notes that shame is a key dramatic element in the *Gorgias* at 461b, 482d-e, 487b, 508b, 494c-499b, 522d. Further, she writes that the word *elenchein* means "to disgrace, put to shame, cross-examine, question, prove, refute, confute, get the better of. The Greek word blurs the distinction between the logical and the psychological, the cognitive and affective dimensions of the experience . . . [E]ach of the three refutations . . . involves shame at a crucial step in the argument" (p. 38).

46    Spitzer (1975) likewise argues that the account of rhetoric as "a branch of the knack of flattery and a sleazy imitation of the true techne . . . is a result of the encounter with Gorgias and a shrewd analysis of the man" (pp. 136–37). In this way, Spitzer argues, the dialogue is "self-referential, demonstrating in *erga* what it argues in discourse" (p. 143).

47    Dodds (1959, p. 225) notes that *kolakeia*, though commonly translated as "flattery", carries a sense in the Greek that is more clearly a slur. The Greek *kolax*, he tells us, finds its equivalent in terms such as "lick-spittle" and "bum-sucker",, and thus *kolakeia* may be translated as "pandering."

48    Dodds (1959, p. 226) notes that Socrates' schematic here reads like the final *diaeresis* of Plato's *Sophist*, according to which sophists and orators are manufacturers of *eidōla* (268b–d).

49    See my interpretation of these other parts of the *Gorgias* in Metcalf (2018). For further analysis of the comedic aspects of the *Gorgias*, see my account of this in Metcalf (2022). Additionally, see Ewegen (2022) and Tanner (2022) in the same volume.

50    See my analysis of Gorgias' *Encomium of Helen* developed at greater length in Metcalf (2022).

51    Long (1998, p. 128) draws parallels between the portrayals of *elenchos* in *Gorgias* and *Theaetetus* and writes that in these texts Plato anticipates the theory of *elenchos* in the *Sophist*: "The modesty that the *elenchos*, or the sophistry of golden lineage, engenders by purgation of persons' 'grand and obstinate opinions concerning themselves' (*Sophist* 230b–d)."

52    Notice that Socrates uses this word in precisely this way in the passage on the *elenchos tou biou* in the *Apology* when he signals to the jury that, after his death, there will be many more who practice *elenchos* who will be harsher than Socrates, and that such a "being-set-free" (*apallangê*) from the reproach that is one's due is neither possible nor honorable (39d). See my interpretation of this in Metcalf (2018).

53    On the philosophical depth of Plato's dialogue-form of writing, see Hyland (1995) and Gordon (1999).

54    See Consigny (2001) for an account of Gorgias' writings as "self-parody": "Gorgias' texts thus mock themselves as well as other texts; and in this respect they are highly self-conscious, self-aware, and self-critical, underscoring their own artificiality and contingency" (p. 176).

55   No doubt the most radical aspect of self-reflexivity in Plato's texts is the critique of (syngrammatic) writing that we find in Plato's *Phaedrus*, *Statesman* and the *Seventh Letter*. See my treatment of this in Metcalf (2017, 2018).

56   Compare Irwin (1986), Consigny (2001) and De Romilly (2002) for their very different perspectives on the differences between Socrates and the sophists.

57   Admittedly, the "agonistic" and "anti-foundationalist" Gorgias presented in Consigny (2001) is less a foil than the Gorgias critiqued by Socrates in Plato's *Gorgias*.

58   An earlier version of this paper was read by Ryan Drake, Shane Ewegen, and Jill Gordon. Many thanks to them, to Michael MacDonald, and to the anonymous reviewers for *Humanities*, for their perceptive critiques of my argument.

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
