# Peer review of "What Performative Contradiction Reveals: Plato’s Theaetetus and Gorgias on Sophistry"

_humanities, doi:10.3390/h12020033_

Round 1

Reviewer 1 Report

There are many excellent insights throughout this paper. My preference would be for a more concise articulation of the insights. I don't myself see why there is a need to supply this amount of Greek, since as far as I can tell there are no discussions of translation or textual variations (I assume the translation is Shorey's Loeb, which if correct should be noted). I don't personally think the many interesting and excellent observations hinge on the extensive textual citation. But these comments are stylistic preferences of mine, and they should be taken as mere suggestions, happily overridden in favor of other styles. The thinking and the specific points about comedy, performative contradiction, and word/deed tension are very interesting. However, I think that it is worth erring on the side of caution, since the sheer volume of translation issues that arise in the citation of the Greek and in the cited passages generally is going to raise questions that, for the most part, I think, distract from the main point of the paper.

It's interesting to note that Protagoras is "ventriloquized", which is noted four times in the introduction, but I don't think it is directly addressed why this is happening, or why it would be relevant to note that it is happening.

Given the focus on doxa and the frequent occurrence of the verb doxazdein, one might expect a discussion of how these terms work in Greek and what their relationship is, but I couldn't find that discussion.

Similarly, pragma and pragmateia are used frequently, and I might expect some setting down of the sense we should expect in these terms, especially given the variation in the English translations offered and the apparent connection with the argumentatively important term "pragmatic".

I'm not sure the Greek citations were consistent, some transliterations were used where one might expect Greek script.

A handful of observations about specific line numbers:

196-197: metrios would be more naturally taken with the verbs that precede it, not with reference to the wisdom of the person whose logos is received.

222-234: Seems this is supposed to be indented.

266: the citation of marginal pagination is inconsistent.

686-687: the translation here is misleading, if it is meant as a translation of the Greek supplied.

Author Response

Reviewer #1 suggests paring back the quantity of Greek text quoted in the manuscript, and qualifies this suggestion as a personal preference possibly to be overridden by others.  I would like to keep the Greek as is in the manuscript, and I see that the Special Issue Editor concurs with my own preference on this point.  In part I believe that the Greek is important precisely because it brings to the fore some of the conceptual connections noted by Reviewer #1—e.g., between doxa and doxazein, between pragma and pragmateia.  I do agree with Reviewer #1 that I note several times the fact that ‘Protagoras’ is ventriloquized in the Theaetetus without explaining the importance of noting that.  Accordingly, I have added to endnote #4 an explanation along the lines that there is—dramatically, at least, if not philosophically—an asymmetry between Socrates going toe-to-toe with Gorgias in Plato’s Gorgias and Socrates refuting his own construction of Protagoras as an interlocutor in Plato’s Theaetetus.  Further, as I make the point in lines 272-273, the ‘Protagoras’ ventriloquized by Socrates in Theaetetus can be seen, philosophically, as something like a combination of Protagoras and Gorgias as sophistic theorists.

Reviewer #1 also observes my citations of Plato’s texts are not always consistent, and I have gone through the manuscript so as to revise these citations to make them consistent.  Reviewer #1 is correct in noting that, in lines 196-197, metriôs should be taken with the verb and not with reference to the wisdom of the person whose logos is received.  I have inserted a comma to make this grammatical construction more evident in the English.  Further, Reviewer #1 notes that the text should be indented for lines 222-234, which is correct.  I did indent that passage, but it does not seem to line up with the other indented passages in the manuscript. Finally, I do see why Reviewer #1 thought that the translation of the Greek supplied at lines 686-687 is misleading, and I revised those lines accordingly. 

Reviewer 2 Report

            This is a beautifully-written and argued piece, to which I offer two things the author might consider and one minor correction. I am grateful for the more nuanced interpretation of sophistry as presented in Platonic dialogues, as too often this becomes read as overly binary and exclusive. Personally, I am inclined to believe that Socrates (and Plato) had more respect for figures such as Protagoras and Gorgias than such interpretations suggest, and in a volume with a theme such as this, this nuance becomes even more significant. The treatment of dialogical form is very thoughtful here.

The first consideration is that I am seeing the extent to which the acknowledgment of performative contradictions illuminates the proximity of sophistry (as Protagoras and Gorgias practice it) and philosophy (as Socrates does). The ‘theater of appearances’ is an interesting point of similarity, but I am not sure whether this constitutes as much of a similarity as the author seems to suggest. The appearances in question are quite different in character, with those on the philosophical side indicating a logos largely independent of them; whereas, those on the sophistic side employ logos merely as a tool, and thus subservient to them. So while I agree with the author that sophists and philosophers may have more in common than often posited, I am not seeing how the author offers a substantial case for the similarities.

The author addresses the Cratylus, and a more theoretical take on the performative contradictions, but Socrates in the Euthydemus makes this explicit and I wonder whether citing this might not strengthen the author’s case. In Plato’s Euthydemus, Socrates makes an overt articulation of the idea of the performative contradiction—specifically, of how the homo mensura argument as Euthydemus and his brother, Dionysodorus, present it ‘upsets itself.’ After Dionysodorus posits the impossibility of contradiction, Socrates says “…I have heard this particular argument from many persons and at many times, and it never ceases to amaze (θαυμαστός) me. The followers of Protagoras made considerable use of it, and so did some still earlier. It always seems to me to have a wonderful way of upsetting not just other arguments, but itself as well (ἀνατρέπων καὶ αὐτὸς αὐτόν)…” (Euthydemus 286c, my emphasis) Where Socrates says that every time he hears this, he is amazed at it, thus humorously undercutting how completely unoriginal and indeed well-worn this claim is (even for Protagoras himself, given the ‘some still earlier’ bit), and how the doctrine upsets itself. As backup to strengthen this case. In Euthydemus, Socrates proceeds to performatively contradicts each of four implications of such a thesis: no false opinion (286d), no such thing as ignorance (286d), no refutation (286e), or mistakes (287a).  Socrates himself performs these contradictions, playing the fool and making mistakes in his counter-argument and then saying “I made a mistake on account of being so stupid.” (287e) 

            Grammatically and stylistically, this has been thoroughly edited, and I find no further editorial matters aside from a minor one in the endnotes. There appear to be extra line spaces that ought to be deleted (or if they are there to differentiate pages to which the notes accord, perhaps this ought to be made clearer).

            I thus submit this review with the strong recommendation that this be published with this minor correction and two more substantive considerations. It is an impressive and important  piece, and a pleasure to read.

Author Response

Reviewer #2 very helpfully notes the parallels with Plato’s Euthydemus 286a-e, and I appreciate the insightful remarks given as to how this passage in Euthydemus can be seen as a confirmation of what is happening with performative contradictions in Theaetetus and Gorgias.  For this reason I have revised endnote #1 to mention this passage in Euthydemus, and I have revised endnote #16 with a more substantial remark as to the significance of Euthydemus 286a-e, particularly with respect to its argument that the homo mensura doctrine would preclude the very possibility of Socratic elenchos.

Reviewer #2 also points out that in some places there are extra line spaces that should be deleted—I have gone through the manuscript to delete these extra line spaces.

Reviewer 3 Report

What Performative Contradiction Reveals: Plato’s Theaetetus and Gorgias on Sophistry

The paper provides an interesting and original analysis of two passages of Plato’s Theaetetus and Gorgias where the philosopher deals with Protagoras’ homo mensura doctrine and Gorgias’ doctrine of the power of logos. As the author points out, in both passages Plato tries to show that these doctrines are each performatively contradicted by the activity of philosophical dialogue. The author persuasively argues that Plato's strategy is, on the one hand, to show the closeness between Socrates and the two sophists, particularly in relation to Socrates’ elenchos, and, on the other hand, to show their distance in their respective conception of our situatedness-within-the-human-world.

The matter discussed in the paper is original, well-argued, and convincing. The paper is clearly written. The bibliography is rich. Perhaps it would be useful to consult the following articles: on the relationship of Protagoras and Socrates, M. Corradi, Protagorean Socrates, Socratic Protagoras: a Narrative Strategy from Aristophanes to Plato, in A. Stavru, C. Moore (eds.), Socrates and the Socratic Dialogue (Leiden-New York 2017), pp. 84-104; on Plato’s analogy between physician and sophist, A. Tordesillas, Protagoras’ Head Exhumed: homo mensura and homo politicus in Plato’s ‘Theaetetus’, in A. Bosch-Veciana, J. Monserrat-Molas (eds.), Philosophy and Dialogue. Studies on Plato’s Dialogues, vol. II (Barcelona 2010), pp. 86-87; on pragmatic aspects of the self-refutation, A. A. Long, Aristotle and the History of Greek Scepticism.” In Studies in Aristotle, edited by D. J. O’Meara,  Washington 1981, 79-106. (= Id., From Epicurus to Epictetus. Studies in Hellenistic and Roman Philosophy, Oxford–New York 2006, 42–69).

In conclusion, I strongly recommend the publication of the article.

Author Response

Reviewer #3 brought to my attention a few articles that are relevant to what I am arguing in my manuscript, and I appreciate these references.  Indeed, I have added Corradi 2018 and Long 2006 to my list of references, and I have revised endnote #9 to include a remark on Long 2006, and endnote #21 to include a remark on Corradi 2018.